SCIENCE FORUM

# A low-cost, open-source evolutionary bioreactor and its educational use

**Abstract** A morbidostat is a bioreactor that uses antibiotics to control the growth of bacteria, making it well-suited for studying the evolution of antibiotic resistance. However, morbidostats are often too expensive to be used in educational settings. Here we present a low-cost morbidostat called the EVolutionary biorEactor (EVE) that can be built by students with minimal engineering and programming experience. We describe how we validated EVE in a real classroom setting by evolving replicate *Escherichia coli* populations under chloramphenicol challenge, thereby enabling students to learn about bacterial growth and antibiotic resistance.

**VISHHVAAN GOPALAKRISHNAN\*[†], DENA CROZIER[†], KYLE J CARD[†], LACY D CHICK, NIKHIL P KRISHNAN, ERIN MCCLURE[‡], JULIA PELESKO, DREW FK WILLIAMSON, DANIEL NICHOL, SOUMYAJIT MANDAL[§], ROBERT A BONOMO AND JACOB G SCOTT\***

**\*For correspondence:**
vxg135@case.edu (VG);
scottj10@ccf.org (JGS)

[†]These authors contributed equally to this work

**Present address:** [‡]University of South Florida Morsani School of Medicine, Tampa, United States; [§]Department of Electrical and Computer Engineering, University of Florida, Gainesville, United States

**Competing interest:** The authors declare that no competing interests exist.

## Introduction

The first bioreactor, the chemostat, was built over 70 years ago to study the growth of bacteria suspended in a liquid medium (*Novick and Szilard, 1950*). In particular, the chemostat maintains the specific growth rate of the bacteria at a fixed value by supplying fresh medium and removing unused nutrients, cells and metabolic byproducts. Studies performed with chemostats and other bioreactors made various physiological and biochemical analyses more tractable and led to the development of quantitative models that describe microbial growth (*Hoskisson and Hobbs, 2005*).

The morbidostat is a bioreactor that maintains a steady death rate. Specifically, a morbidostat regulates the bacterial growth rate by measuring the cell density of a population at fixed time intervals and subsequently introducing factors that cause the growth rate to increase (such as nutrient-rich media) or factors that cause the growth rate to decrease (such as antibiotic-rich media). Antibiotics are introduced into the medium when the population size exceeds a predetermined threshold and the growth rate is greater than zero. The morbidostat, therefore, adjusts the selective pressure to maintain near-constant growth inhibition as the population becomes resistant to the antibiotics over time. Studies performed with morbidostats have improved our understanding of the evolution of antibiotic resistance in bacteria (*Toprak et al., 2012*; *Dößelmann et al., 2017*; *Yoshida et al., 2017*; *Leyn et al., 2021*) and the temperature-stress response in yeast (*Wong et al., 2018*).

Introducing the morbidostat into high-school or college curricula has many benefits. First, educators often use passive, lecture-based instruction to teach gradual long-term species evolution. In contrast, in-classroom morbidostat use enables inquiry-based research where students combine hypothesis generation and experimental design with measurement of real-time adaptation of microbial populations. This active learning approach improves student understanding of bacterial growth and how natural selection acts on heritable genetic variation to promote drug resistance (*Freeman et al., 2014*; *Tomandl et al., 2015*). In the context of biology classes, the use of a morbidostat in the classroom complements existing inquiry-based laboratory curricula (*Cooper et al., 2019*). Second, constructing a morbidostat will teach students

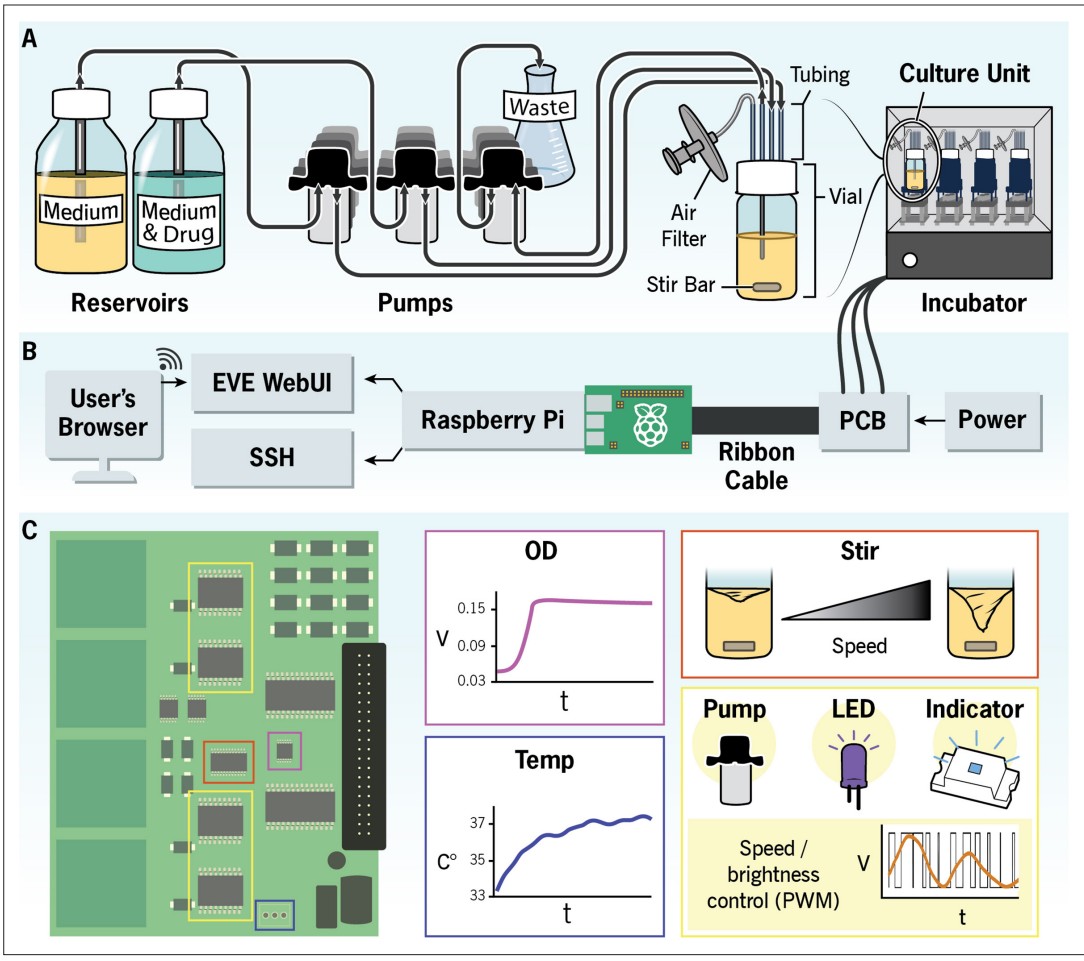

**Figure 1.** The EVolutionary bioreactor (EVE). Schematic illustration of the EVE hardware and software architecture. (**A**) Reservoirs containing permissive (i.e., drug-free) and selective growth media are each connected to a culture unit (CU) through silicone tubing. Peristaltic pumps control the rate at which the media are added to the evolving bacterial population, and an additional pump removes waste. Multiple CUs can fit into an incubator, allowing users to simultaneously evolve several independent replicate populations. (**B**) A Raspberry Pi interfaces with a custom printed circuit board (PCB) to monitor culture growth and coordinate the addition and removal of growth media and waste. The user controls the EVE hardware with a web application. (**C**) The PCB chips measure voltage and temperature. These voltage measurements are proportional to optical density and thus are a way to estimate population size. The chips also control the fluidic pumps, diodes, onboard indicators, and stirring speed. The PCB diagram illustrates where these chips are located on the board.

critical engineering and computer programming skills.

We have developed a low-cost open-source morbidostat called the EVolutionary biorEactor (EVE) with these goals in mind. Our platform, which builds on the methodology outlined by Toprak and colleagues (*Toprak et al., 2012*; *Toprak et al., 2013*), supports multiple culture units that each run independently with precise control over exchange of the growth medium, cell density measurement rates, and drug addition. EVE costs about $115–200 to construct using printed circuit board (PCB) and 3D-printed component files; circuit diagrams, software and complete build instructions are available in our GitHub repository (*Gopalakrishnan, 2022*).

In this article we discuss the design and construction of EVE. We also describe how high school students in a class taught by one of the authors (LDC) used it to study the growth of bacteria and the evolution of antibiotic resistance, demonstrating that EVE is an accessible alternative to other bioreactors for use in low-resource classrooms, and in settings where instructors and students have limited engineering and programming experience.

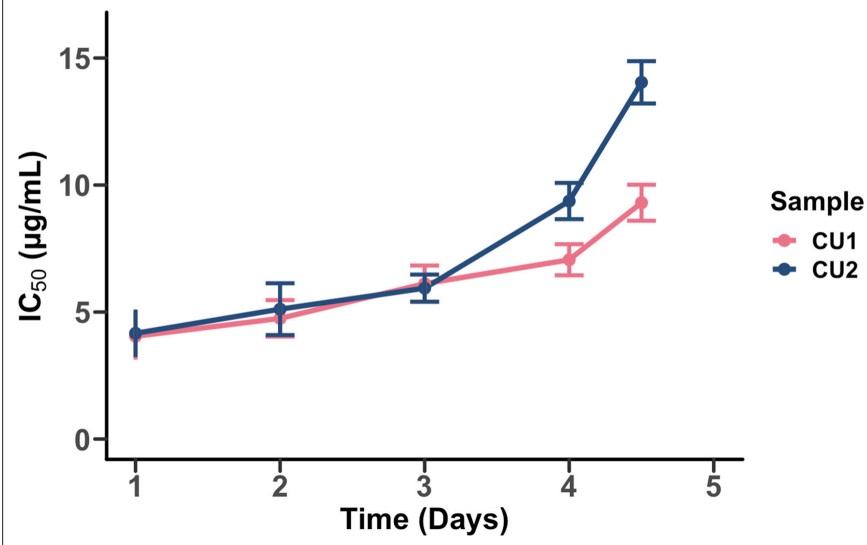

**Figure 2.** The evolution of chloramphenicol resistance over time. Half-maximal inhibitory concentration ($IC_{50}$) values for two biologically independent replicates (CU1 and CU2) are plotted against time. Points represent the estimated $IC_{50}$ determined by fitting a Hill function to $OD_{600}$ measurements across this dilution series after 18 hours of population growth. Error bars represent the variance in $IC_{50}$. The data used to generate this figure are available in our Github repository: https://github.com/vishhvaan/eve-pi (*Gopalakrishnan, 2022*).

## The EVolutionary biorEactor

The EVE is functionally similar to other small, automated bioreactors (*Toprak et al., 2012*; *Dößelmann et al., 2017*; *Wong et al., 2018*). In summary, several bacterial cultures are grown simultaneously in small glass vials under well-mixed, drug-selective conditions (*Figure 1A*). A control algorithm maintains these conditions by: (i) supplying power to fans (with attached magnets) that rotate magnetic stir bars; (ii) introducing fresh medium into each culture at a constant rate through the activation of peristaltic pumps; (iii) monitoring population growth via absorbance measurements taken from paired light-emitting and photo-sensitive diodes; (iv) introducing selective medium when these absorbance measurements exceed a user-defined value. Nonetheless, the EVE differs from other bioreactors in its hardware, software, and production costs. We detail these aspects below.

### Hardware and software

The EVE uses a primary onboard Raspberry Pi microcomputer to execute the software and serve as a bridge between the user interface and a PCB (*Figure 1B, C*). This design keeps costs low while allowing the device to be self-contained in environments without internet access. One can also construct the EVE with a breadboard. Breadboards give users more flexibility to modify

the hardware but require prior experience and come at the cost of increased assembly time and device footprint. In either case, circuit diagrams, 3D-printed component files, complete build instructions, and a part list are in our GitHub repository (*Gopalakrishnan, 2022*).

One may access population growth and temperature measurements, edit configuration files, define experimental parameters, and monitor and control experiments remotely with our free, open-source software run inside any web browser. Users install this software by downloading the pre-compiled Raspberry Pi image or executing the installation script directly from our GitHub repository. Data can be saved to network locations mounted to the Pi's file system or an attached USB device.

### Comparisons to alternative bioreactors

The EVE differs from other bioreactors, including the eVOLVER (*Wong et al., 2018*) and Flexostat (*Takahashi et al., 2014*), in its design philosophy and customization capabilities. The EVE and eVOLVER software are hosted on a Raspberry Pi and written in Python, a common programming language with a broad user base and community support. This combination of hardware and software allows for fast code execution and the Linux operating system facilitates customization. In contrast, other automated culture systems use proprietary or third-party software that may be inaccessible or cost-prohibitive for educators (*Toprak et al., 2012*).

One does not need to be familiar with Python to install the software, edit experimental parameters, or run the pre-configured control algorithms. However, working knowledge of this language is necessary to create custom algorithms, for which we uniquely provide design instructions in our GitHub repository. Moreover, the EVE uses a more modern serial connection between the motherboard and the Pi than the eVOLVER, which permits the use of more up-to-date software packages. Lastly, the Flexostat and the EVE have a similar broadly open software license that permits unrestricted use of the software.

### Production cost

We designed the EVE to be cost-effective. The total cost varies between $115 and $200 to build a system that performs simple growth or evolution experiments in triplicate with a negative control. These estimates include all parts except the incubator, glassware, and 3D printer. The circuit board design can be downloaded from our

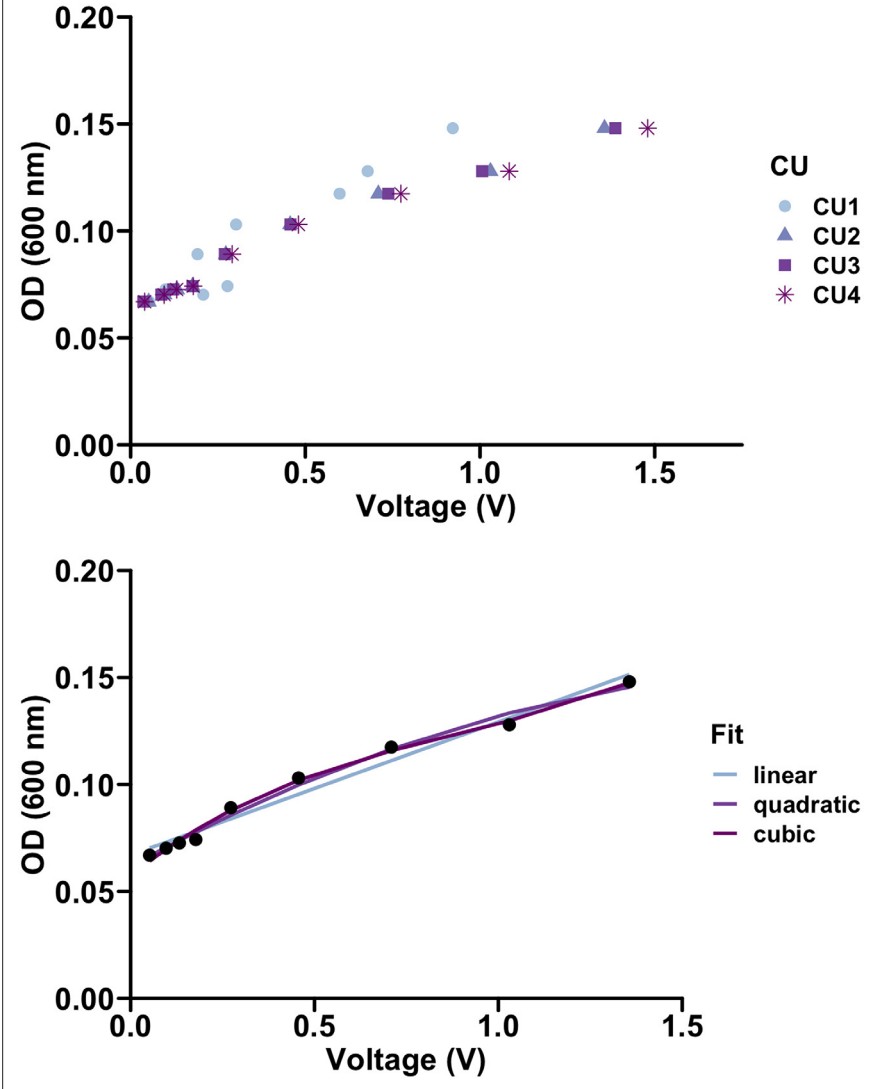

**Figure 3.** Calibration of culture optical density with reported voltage. (Top) Culture units behave similarly when measuring voltage. (Bottom) Various regression methods were used to quantify the relationship between the optical density (OD) and voltage. $R^2$ values of 0.97, 0.99, and 0.99 were calculated for the linear, quadratic, and cubic fits respectively. We used a linear model to translate voltage to optical density in our experiments.

these results with previously reported measurements (*Toprak et al., 2012*). Voltage is the raw measurement that is used to estimate optical density (i.e., population size). Smaller voltage measurement variability between individual culture units indicates better construct and internal validity. Variability in measurements is introduced due to subtle differences when printing the vial holder, leading to slightly altered positioning of the diodes responsible for voltage measurements. The average variability was 8.0% (mean ± sd = 2.93±0.23), comparable to the 7.5% of Toprak and colleagues.

Second, we experimentally evolved bacterial populations and again compared the results to those of *Toprak et al., 2012*. We revived *E. coli* ATCC 25922 from a frozen stock by overnight growth in Mueller-Hinton broth (MHB) (Sigma). We inoculated 100 µL of this culture into three separate culture units containing 12 mL of MHB. We then prepared the selective growth medium by diluting a chloramphenicol stock solution to 40 µg/mL in MHB. The selective and permissive media were connected to the culture units, which were incubated at 37 °C with constant stirring (~225 rpm). Samples were taken from each culture's effluent waste every 12 hours and frozen at –80 °C with 15% glycerol as a cryoprotectant.

We estimated each sample's half-maximal inhibitory concentration ($IC_{50}$) following the broth microdilution method (*Wiegand et al., 2008*) and by fitting a Hill function to the resulting optical density data (*Maltas and Wood, 2019*). The three ancestral populations began the experiment with an $IC_{50}$ of 4 µg/mL, and chloramphenicol resistance increased to 9.3–13.9 µg/mL. These results are comparable to *Toprak et al., 2012*, where resistance increased, on average, to approximately 15 µg/mL over the same time period (*Figure 2*).

repository and sent to a PCB manufacturer for printing and assembly; in our case, we purchased fully assembled PCBs for approximately $42 per board, including the power supply. Users may further decrease costs and increase accessibility in low-resource classrooms by incubating bacterial cultures at room temperature and using a pressure cooker to sterilize growth media instead of an autoclave.

### *Validation*

We validated the EVE in two ways. First, we examined voltage measurement variability across seven independent culture units and compared

## Educational use

The EVE is well-suited for classroom settings because it was designed in collaboration with educators and tested by high school students at Hawken School in Gates Mills, Ohio. We discuss our curriculum design, the students' experience using the EVE, and its potential applications in other educational contexts.

We first examined the Advanced Placement (AP) Biology curriculum and then developed a pilot curriculum for the evolutionary unit of the course that detailed the desired learning outcomes, assessment evidence for those outcomes, and a learning plan (see *Supplementary file 1*). The

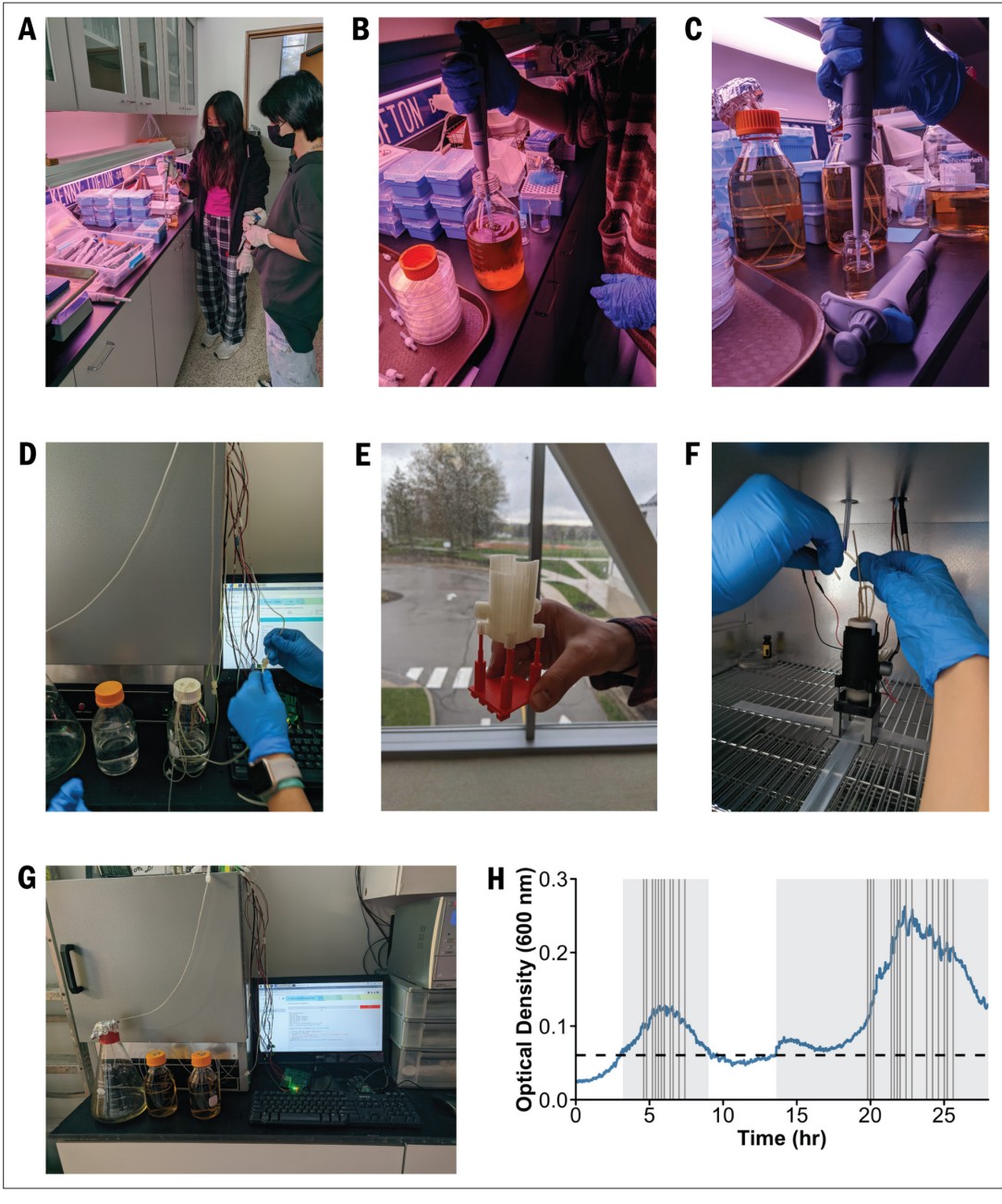

**Figure 4.** High school students at Hawken School built the EVE and performed an evolution experiment. (**A**) Prior to experimentation, the students practiced pipetting with media. (**B**) They prepared the selective growth medium and (**C**) inoculated *E. coli* into a culture unit containing the permissive medium. (**D**) The system was sterilized with bleach and ethanol solutions. (**E**) Before the experiment, the students printed a culture unit stand in the makerspace at Hawken School. (**F**) The media and waste reservoirs were attached to the culture unit by silicon tubing, and then (**G**) the students began the experiment. (**H**) Bacterial growth and inhibition in the EVE over 28 hours. The grey boxes and vertical lines indicate when the EVE control algorithm added the permissive and selective media into the culture units, respectively.

AP curriculum introduces several key learning objectives about the importance of phenotypic variation, how natural selection acts on this variation, and how this phenomenon affects populations over time. The AP instructional model also emphasizes that students use supporting resources and perform appropriate experiments to build and strengthen their conceptual understanding of these objectives. More generally, for most biology classrooms, bioreactor experiments introduce basic microbiology techniques, biotechnology, and data analysis to students.

## Box 1. Testimonials from students.

The bioreactor experiment was a very valuable learning experience. Learning the evolution of bacterial growth against drug resistance deepened my understanding of the wonders of the evolution unit in my current biology class. Through our hands-on experience, I have learned many new terms and biotechnology techniques including pipetting, working with bacterial cultures, and learning the Beer's Law. This was a complex but also at the same time simple lab to do, and I think it would definitely be beneficial for students to learn more about bacterial evolution and practice important skills such as calculating concentrations and analyzing data.
— Lillian Fu.

The Bioreactor experiment was very informational for me; I learned how bacterial growth works and how drug resistance plays a role in it through my first-hand experience with being able to actually work with the bacterial cultures, drugs, pipettes, and EVE. I think this was a fun project to do, and it would definitely be an easy and helpful project for other high school students to do while learning about bacterial growth and evolution. Students can also utilize and strengthen skills such as analyzing data and calculating different concentrations. Overall, I think this project was accessible and simple but still very interesting and informative.
— Grace Shum.

We therefore developed our unit to facilitate student understanding of population growth dynamics and trait evolution through independent experimentation with the EVE. To reinforce this objective, we asked the students to consider how bacterial populations respond to changing environments, such as introducing antimicrobial agents.

Two high school students followed the instructions in our GitHub repository (*Gopalakrishnan, 2022*) to assemble the EVE bioreactor with 3D-printed equipment, a fabricated circuit board, and a Raspberry Pi microcomputer with the necessary pre-installed software. Then two AP Biology students, working as a team, calibrated the device (*Figure 3*) and evolved a population of *E. coli* ATCC 25922 to increasing chloramphenicol concentrations over several days (*Figure 4A–G*). The population initially expanded in size, decreased after the introduction of the antibiotic into the culture unit, and later rebounded as drug-resistant variants rose in frequency (*Figure 4H*). From these data, the students calculated growth rates and predicted how these rates would change with varying drug selective pressures. This authentic research experience introduced the students to standard microbiology practices and concepts, including how to prepare drug solutions and use sterile technique to maintain bacterial cultures, when logistic growth models apply, and the relationship between light absorbance and cell number (i.e., Beer's law). We share testimonials from these students in *Box 1*.

Notably, the students adjusted the experimental protocol to account for classroom limitations. For example, the Hawken School does not have Bunsen burners or standing incubators. The students therefore created a spartan workspace and disinfected all surfaces with 70% ethanol, visually inspected the growth medium to ensure that it remained free of contamination, and incubated bacterial cultures in an oven set to approximately 30–40°C. Moreover, although this particular high school has autoclaves, there are several alternative ways that users can sterilize glassware and media, including using liquid chemicals and microwaves. The Centers for Disease Control and Prevention describe some of these methods in greater detail (https://www.cdc.gov/infectioncontrol/guidelines/disinfection/sterilization/other-methods.html).

One could also use the EVE in other educational contexts (i.e., college laboratory classrooms) or outside the classroom. Since the EVE functions through a combination of engineering and biology, project work within clubs may offer unique experiences to students wishing to learn about how multiple disciplines interact. These projects could combine construction and experimentation for a seamless educational experience. Interested parties may acquire non-pathogenic *E. coli* K12, growth medium, and antibiotic powders through the Carolina Biological Supply Company or a similar vendor.

The EVE has also been implemented in several other contexts. For instance, the EVE is being used to study bacteriophages in continuous

culture at the University of Exeter in the United Kingdom, and to develop research bioreactors at a French biotech company. Additionally, undergraduate students used its design to build their own custom morbidostat as part of the International Genetically Engineered Machine (iGEM) competition. The manual produced by these students represents an example of what the EVE would look like in a college setting (https://static.igem.org/mediawiki/2019/c/c6/T--Athens--Morbidostat_manual.pdf).

## Future directions and conclusion

In addition to its educational utility, the EVE can address questions of evolutionary repeatability. For instance, one might examine whether correlated drug responses are conserved across time as populations evolve under single- or multi-drug selection (*Nichol et al., 2015*; *Nichol et al., 2019*; *Card et al., 2021*; *Iram et al., 2021*; *King, 2022*). Although the current EVE system can only introduce one drug solution into the growth medium, we are designing a PCB to allow the simultaneous or sequential addition of multiple drugs across more replicate cultures. Moreover, users could substitute the existing hardware with LEDs and photodiodes corresponding to fluorescence proteins' excitation and emission frequencies. This hardware alteration would allow bacterial head-to-head competitions without periodic sampling and cell enumeration.

The EVE device has several limitations, some of which we will address with additional design and hardware improvements. Although one could buy 3D-printed parts from several internet vendors, the need of a 3D printer may still restrict use in resource-limited classrooms. Future implementations will include alternative hardware construction methods that preclude the need for a 3D printer and thus lower indirect costs. Second, the EVE may require slightly more setup than a pre-constructed and calibrated commercially available bioreactor. For example, fluid pumps may vary in flow rate; thus, they must be individually calibrated to avoid accidental overflow. Although there is educational value in solving hardware and software challenges, especially in a classroom setting, we encourage individuals to use EVE's GitHub repository to report challenges and potential solutions. We will also continue to work on automated solutions to mitigate these setup tasks for the user.

In summary, we believe that EVE is uniquely suited for use in educational settings because of its low-cost and open-source design. High school students demonstrated these aspects by building the EVE and performing a simple evolution experiment in the classroom. We are optimistic about a future where more classrooms use the EVE and other platforms to spur student interest in experimental biology and technology.

**Vishhvaan Gopalakrishnan** is in the Lerner College of Medicine, Cleveland Clinic, Cleveland, United States

vxg135@case.edu

http://orcid.org/0000-0002-0532-7710

**Dena Crozier** is in the Lerner College of Medicine, Cleveland Clinic, and the Case Western Reserve University School of Medicine, Cleveland, United States

http://orcid.org/0000-0002-1841-0011

**Kyle J Card** is in the Department of Translational Hematology and Oncology Research, Cleveland Clinic, Cleveland, United States

http://orcid.org/0000-0002-0462-2777

**Lacy D Chick** is at Hawken School, Gates Mills, United States

http://orcid.org/0000-0002-3059-4300

**Nikhil P Krishnan** is at Case Western Reserve University School of Medicine, Cleveland, United States

**Erin McClure** is in the Department of Translational Hematology and Oncology, Research, Cleveland Clinic, Cleveland, United States

http://orcid.org/0000-0002-6604-1273

**Julia Pelesko** is in the Department of Translational Hematology and Oncology Research, Cleveland Clinic, and the Department of Physics, Case Western Reserve University, Cleveland, United States

**Drew FK Williamson** is in the Department of Pathology, Massachusetts General Hospital, Boston, the Department of Pathology, Brigham & Women's Hospital, Boston, and the Cancer Program, Broad Institute of Harvard and MIT, Cambridge, United States

http://orcid.org/0000-0003-1745-8846

**Daniel Nichol** is in the Centre for Evolution and Cancer, Institute of Cancer Research, London, United Kingdom

http://orcid.org/0000-0003-2662-1836

**Soumyajit Mandal** is in the Integrated Circuits and Sensor Physics Lab, Case Western Reserve University School of Engineering, Cleveland, United States

**Robert A Bonomo** is in the Louis Stokes Cleveland Department of Veteran Affairs Medical Center, Cleveland, United States; the Departments of Medicine, Pharmacology, Molecular Biology and Microbiology, Biochemistry, and Proteomics and Bioinformatics, Case Western Reserve University School of Medicine, Cleveland, United States; and the CWRU-Cleveland VAMC Center for Antimicrobial Resistance and Epidemiology, Cleveland, United States

**Jacob G Scott** is in the Department of Translational Hematology and Oncology Research, Cleveland Clinic,

the Department of Physics, Case Western Reserve University, and the Department of Radiation Oncology, Cleveland Clinic, Cleveland, United States
scottj10@ccf.org
 http://orcid.org/0000-0003-2971-7673

*Author contributions:* Vishhvaan Gopalakrishnan, Conceptualization, Resources, Data curation, Software, Formal analysis, Supervision, Validation, Investigation, Visualization, Methodology, Writing – original draft, Project administration, Writing – review and editing; Dena Crozier, Formal analysis, Validation, Investigation, Visualization, Writing – review and editing; Kyle J Card, Formal analysis, Supervision, Validation, Investigation, Visualization, Methodology, Writing – review and editing; Lacy D Chick, Resources, Methodology; Nikhil P Krishnan, Conceptualization, Software, Formal analysis, Investigation; Erin McClure, Validation, Investigation; Julia Pelesko, Investigation; Drew FK Williamson, Conceptualization, Software, Investigation; Daniel Nichol, Conceptualization; Soumyajit Mandal, Methodology; Robert A Bonomo, Conceptualization; Jacob G Scott, Conceptualization, Resources, Supervision, Funding acquisition, Validation, Investigation, Visualization, Methodology, Project administration, Writing – review and editing

*Competing interests:* The authors declare that no competing interests exist.

## Funding
No external funding was received for this work.

## Decision letter and Author response
Decision letter https://doi.org/10.7554/eLife.83067.sa1
Author response https://doi.org/10.7554/eLife.83067.sa2

## Additional files

### Supplementary files
• Supplementary file 1. Curriculum unit plan.
• MDAR checklist

### Data availability
We provide the materials to build an EVE in our Github repository: https://github.com/vishhvaan/eve-pi (copy archived at swh:1:rev:92c8be59379f42b-f9e210270704456b1d02f0e44). We have also included the data to generate Figure 2 on the Github.

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
