## [Decision Letter]

**Decision letter after peer review:**

Thank you for submitting your article "A Low-Cost, Open Source, Self-Contained Bacterial EVolutionary biorEactor (EVE)" for consideration by *eLife*. Your article has been reviewed by 3 peer reviewers, and the evaluation has been overseen by a Reviewing Editor and Patricia Wittkopp as the Senior Editor. The following individual involved in review of your submission has agreed to reveal their identity: Vaughn S Cooper (Reviewer #1).

The reviewers have discussed the reviews with one another and the Reviewing Editor has drafted this decision in consultation with the Senior Editor.

The editors have judged that your manuscript is of interest, but as described below much additional work is needed to make it suitable for publication in *eLife*. We suspect that these changes will require more than the 2 months typically allowed for revisions, so are rejecting the paper in its current form. We are, however, very interested in seeing a revised version that addresses the reviewers concerns. In fact, the journal is currently considering adding a new category to its Features section focusing on education and outreach, and we think this work would be a good fit for that format. We share the author's passion for finding ways to bring experimental systems into high-schools and think that the changes requested below will make adoption more likely.

Summary:

This article introduces a flexible continuous culture device built with open source hardware and based on readily (and cheaply) available hardware. The paper also presents a proof of concept using antibiotic selection to demonstrate the ability to use feedback control. The manuscript claims it can replicate the functionality of more elaborate and expensive reactors on the market and thus enable students to build their own devices and conduct their own research. However, no such data are presented to substantiate these claims, either to demonstrate the performance of the device relative to the competing standards or to show its usability by less trained students.

Essential revisions:

1) Provide evidence for the first claim: present data that shows that the bioreactor works and provides useable data and compare the bioreactor's result to other currently available reactors. While we don't expect that the authors purchase expensive reactors for these comparisons themselves, they should compare their findings with the product specs of others. How variable are the replicates, typically? What are key issues to be cautious with? Are the parts easy to come by, even though they are cheap?

2) Show ease of use, e.g. by providing the parts of a bioreactor together with a manual to a number of college or high school classes, and ask them to build the reactor and perform a simple experiment. They could then report the data on how many classes were able to do this, and how the data compares with a reference experiment. Even several anecdotal examples of it being used in non-traditional lab settings? The more of these the better! Please also add a photograph of the device.

3) Regarding educational use -- what would they learn? What are the learning objectives for experiments based on this setup?

4) Who is the target user, and what must they know to succeed?

5) What kinds of experiments would this target user ideally conduct?

6) We would like to know more specifically how this method advances your research, and think a research advance (rather than Amp killing E coli) would be the use case that proves its merit.

7) Figure 2 is hard to follow, and without some effort. The letter labels are placed oddly for my preference, a border around the subfigures might make that easier. It took a little bit of time to figure out what I was looking at in b) with regard to the concentration and the voltage. I also didn't see the data from b) in c) but was expecting to.

8) There is no data that suggests that drug concentrations can be set to a desired and reliable level. Please provide.

[Editors' note: further revisions were suggested prior to acceptance, as described below.]

Thank you for submitting the revised version of your article "A low-cost, open-source evolutionary bioreactor and its educational use" to *eLife* for consideration as a Feature Article. I sent the revised version of the article to the three reviewers who reviewed the original version and their comments are below. As you will see, there are still a small number of points you need to address. The following individuals involved in review of your submission have agreed to reveal their identity: Vaughn Cooper.

Summary

This paper presents the author's low-cost continuous culture device EVE. It presents a case study of this device as a morbidostat, and its use within a high school AP Biology lesson. The stated goal is to show how this open source device is a viable alternative to other more expensive or cumbersome technology. This is a great idea and the method is impressive. However, there are a number of points that need to be addressed to make the article suitable for publication.

Essential revisions:

1. Evidence for the efficacy of the program is anecdotal: how well did students learn AP material?

2. Please say more about the challenges of implementation and any potential limitations of this approach/device.

3. Please say more about other options (e.g., eVOLVER, Flexostat, commercial bioreactors) and how EVE compares to them. What are the benefits and/or unique features of EVE compared with other options.

---

## [Author Response]

Summary:This article introduces a flexible continuous culture device built with open source hardware and based on readily (and cheaply) available hardware. The paper also presents a proof of concept using antibiotic selection to demonstrate the ability to use feedback control. The manuscript claims it can replicate the functionality of more elaborate and expensive reactors on the market and thus enable students to build their own devices and conduct their own research. However, no such data are presented to substantiate these claims, either to demonstrate the performance of the device relative to the competing standards or to show its usability by less trained students.

We thank the reviewers for raising this point. We have now addressed this concern in two ways. First, we replicated an experiment from Toprak et al. (2012) by challenging replicate *E. coli* populations to increasing chloramphenicol concentrations over time. We detail this work and our confirmatory results in a new Validation section. Second, we introduced the EVE into a high school classroom setting. The students constructed the device from available parts and experimentally evolved a bacterial population. We have now included a new section (Educational Use) detailing their experience.

Essential revisions:1) Provide evidence for the first claim: present data that shows that the bioreactor works and provides useable data and compare the bioreactor's result to other currently available reactors. While we don't expect that the authors purchase expensive reactors for these comparisons themselves, they should compare their findings with the product specs of others. How variable are the replicates, typically? What are key issues to be cautious with? Are the parts easy to come by, even though they are cheap?

To address this concern, we replicated an experiment from Toprak et al. 2011. We used the EVE to evolve three replicate populations over 4.5 days to dynamically increasing chloramphenicol concentrations. Our results were qualitatively similar to those in Figure 2 of their paper. For instance, the magnitude of resistance increases from the drug-susceptible *E. coli* ancestors (MG1655 in their case and ATCC 25922 in ours) was roughly the same over the same period. We report our results in the new Validation section.

2) Show ease of use, e.g. by providing the parts of a bioreactor together with a manual to a number of college or high school classes, and ask them to build the reactor and perform a simple experiment. They could then report the data on how many classes were able to do this, and how the data compares with a reference experiment. Even several anecdotal examples of it being used in non-traditional lab settings? The more of these the better! Please also add a photograph of the device.

We thank the reviewers for this suggestion and have reframed the manuscript to emphasize the EVE’s educational utility. We provided hardware and instructions to high school students and asked them to build the EVE and perform simple growth and evolution experiments. We detail these efforts in our new Educational Use section, as follows (page 5):

“Two high school students followed the instructions in our GitHub repository to assemble the EVE bioreactor with 3D-printed equipment, a fabricated circuit board, and a Raspberry π microcomputer with the necessary pre-installed software. Then three AP Biology students, working as a team, calibrated the device and evolved a population of *E. coli* ATCC 25922 to increasing chloramphenicol concentrations over several days.”

Moreover, we have included several examples of EVE’s use in other non-traditional settings (page 7):

“The EVE has been implemented in several laboratories and classrooms throughout the world. For instance, the EVE is being used to study bacteriophages in continuous culture at the University of Exeter, and used to develop research bioreactors in a French biotech company. Undergraduate students used its designs to build their own custom morbidostat as part of the International Genetically Engineered Machine (iGEM) competition. Their manual represents an example of what the EVE would look like in a college setting.”

Lastly, we added a schematic illustration of the device (Figure 1) and photos of the high school experimental setup (Figure 3).

3) Regarding educational use -- what would they learn? What are the learning objectives for experiments based on this setup?

We collaborated with Dr. Lacy Chick (now a co-author on our revised manuscript) to address this issue. Dr. Chick is a high school biology teacher at the Hawken School in Gates Mills, Ohio. She has experience in curriculum design, including developing unit plans for AP and general biology courses. With her expertise, we co-developed a unit plan with essential questions and goals centered on the EVE. These documents are now included in the supplementary materials. Moreover, we added several sentences to the Educational Use section (page 5):

“We first examined the Advanced Placement (AP) Biology curriculum and developed a unit plan to complement this program. The AP curriculum introduces several key learning objectives about the importance of phenotypic variation, how natural selection acts on this variation, and how this phenomenon affects populations over time. The AP instructional model also emphasizes that students use supporting resources, and perform appropriate experiments, to build and strengthen their conceptual understanding of these objectives. More generally, for most biology classrooms, experiments with a bioreactor introduce basic microbiology techniques, biotechnology, and data analysis.”

4) Who is the target user, and what must they know to succeed?

We are targeting academic users and students. In the manuscript we present possibilities for both groups. Instructions and materials to construct the device are in our Github repository.

5) What kinds of experiments would this target user ideally conduct?

We now include a Future Directions section that highlights two possibilities (page 7):

“In addition to its educational utility, the EVE is an ideal system to address questions of evolutionary repeatability. For instance, one might examine whether correlated drug responses are conserved across time as populations evolve under single- or multi-drug selection. Although the current EVE system can only introduce one drug solution into the growth medium, we are currently designing a PCB that will allow the simultaneous or sequential addition of multiple drugs. Second, users could substitute the existing hardware with LEDs and photodiodes corresponding to fluorescence proteins’ excitation and emission frequencies. This hardware alteration would allow bacterial head-to-head competitions without periodic sampling and cell enumeration.”

Students could perform evolution experiments, as was done in this manuscript, or even simple bacterial growth experiments.

6) We would like to know more specifically how this method advances your research, and think a research advance (rather than Amp killing E coli) would be the use case that proves its merit.

We now focus on the EVE’s educational value, and therefore we place less emphasis on how the device advances our research. However, we include two possible future studies that our device would be well-suited for.

7) Figure 2 is hard to follow, and without some effort. The letter labels are placed oddly for my preference, a border around the subfigures might make that easier. It took a little bit of time to figure out what I was looking at in b) with regard to the concentration and the voltage. I also didn't see the data from b) in c) but was expecting to.

We have removed this figure from the manuscript.

8) There is no data that suggests that drug concentrations can be set to a desired and reliable level. Please provide.

We have removed this claim.

[Editors' note: further revisions were suggested prior to acceptance, as described below.]

Essential revisions:1. Evidence for the efficacy of the program is anecdotal: how well did students learn AP material?

Because this pilot program was a special project within the overarching AP curriculum, two students constructed and used the EVE to perform the evolution experiment. Therefore, we decided to focus on the students’ qualitative experiences through testimonials, as formal statistics on a small sample size would be less meaningful. Nevertheless, we agree that collecting quantitative data which compares student performance on evolution-related test questions or qualitative pre- and post-experiment surveys evaluating self-reported learning would be invaluable outcomes to gather in the future. As we perform further investigations on how the EVE can fit into bigger and more diverse classrooms, we will be sure to gather this outcome data.

2. Please say more about the challenges of implementation and any potential limitations of this approach/device.

We have added this paragraph to the *Future Directions* section to address limitations:

“The EVE device has several limitations, some of which we will address with additional design and hardware improvements. Although one could buy 3Dprinted parts from several internet vendors, the need of a 3D printer may still restrict EVE’s use in resource-limited classrooms. Future implementations will include alternative hardware construction methods that preclude the need for a 3D printer and thus lower indirect costs. Second, the EVE may require slightly more setup than a pre-constructed and calibrated commercially available bioreactor. For example, fluid pumps may vary in flow rate; thus, they must be individually calibrated to avoid accidental overflow. Although there is educational value in solving hardware and software challenges, especially in a classroom setting, we encourage individuals to use EVE’s GitHub repository to report challenges and potential solutions. We will also continue to work on automated solutions to mitigate these setup tasks for the user.”

3. Please say more about other options (e.g., eVOLVER, Flexostat, commercial bioreactors) and how EVE compares to them. What are the benefits and/or unique features of EVE compared with other options.

We added a section called Comparisons to Alternative Bioreactors under the Hardware and Software section. This section addresses alternative bioreactors and where the EVE differs:

“The EVE differs from other bioreactors, including the eVOLVER and Flexostat, in its design philosophy and customization capabilities. The EVE and eVOLVER software are written in Python, a common programming language with a broad user base and community support, and hosted on a Raspberry π microcomputer. This combination of hardware and software allows for fast code execution and a Linux operating system that enables customization. In contrast, other automated culture systems use proprietary or third-party software that may be inaccessible or cost-prohibitive for educators. One does not need to be familiar with Python to install the software, edit experimental parameters, or run the pre-configured control algorithms. However, working knowledge of this language is necessary to create custom algorithms, for which we uniquely provide design instructions in our GitHub repository. Moreover, the EVE uses a more modern serial connection between the motherboard and the π than the eVOLVER, which permits the use of more up-to-date software packages. Lastly, the Flexostat and the EVE have a similar broadly open software license that permits unrestricted use of the software.”